# Effects of Game-Specific Demands on Accelerations during Change of Direction Movements: Analysis of Youth Female Soccer

Aki-Matti Alanen [1,2,*], Lauren C. Benson [2,3], Matthew J. Jordan [1,4], Reed Ferber [4,5,6] and Kati Pasanen [1,2,4,7,8,9]

1 Integrative Neuromuscular Sport Performance Laboratory, Faculty of Kinesiology, University of Calgary, Calgary, AB T2N 1N4, Canada; mjordan@ucalgary.ca (M.J.J.); kati.pasanen@ucalgary.ca (K.P.)
2 Sport Injury Prevention Research Center, Faculty of Kinesiology, University of Calgary, Calgary, AB T2N 1N4, Canada; lauren.benson@ucalgary.ca
3 Tonal Strength Institute, Tonal, San Francisco, CA 94107, USA
4 Faculty of Kinesiology, University of Calgary, Calgary, AB T2N 1N4, Canada; rferber@ucalgary.ca
5 Running Injury Clinic, Calgary, AB T2N 1N4, Canada
6 Faculty of Nursing, Cumming School of Medicine, University of Calgary, Calgary, AB T2N 1N4, Canada
7 McCaig Institute for Bone and Joint Health, University of Calgary, Calgary, AB T2N 1N4, Canada
8 Alberta Children's Hospital Research Institute, University of Calgary, Calgary, AB T2N 1N4, Canada
9 Tampere Research Center of Sports Medicine, UKK Institute, 33500 Tampere, Finland
* Correspondence: akimatti.alanen@ucalgary.ca

**Abstract:** The aim of this study was to assess center of mass (COM) acceleration and movement during change of direction (COD) maneuvers during a competitive soccer game to elucidate situation-specific demands of COD performance. This information can assist in developing soccer-specific tests and training methods. Fifteen elite-level female youth soccer players were tracked for one game with inertial measurement units (IMU) attached to the lower back. COD movements in combination with situational patterns were identified using high-speed video. LASSO regression was used to identify the most important predictors associated with higher vertical peak accelerations ($PA_v$) of the COM during COD movements. COD angle, running speed, contact, and challenge from the opposition were identified as important features related to higher $PA_v$. This study adds to the literature on the demands of COD performance in soccer match-play. The unique approach with game-specific situational data from female youth players provides increased insight into the game-demands of COD and agility performance. $PA_v$ in games was higher with larger COD angles, increased running speed, or with contact when the player was challenged by the opposition. A larger study including more games is warranted to increase confidence in using these variables as a basis for training or testing agility.

**Keywords:** football; soccer; change of direction; game analysis; situational patterns



## 1. Introduction

Change of direction (COD) performance is an important discriminator of elite players in youth soccer and is also associated with common mechanisms of injury [1–4]. Common COD tests in youth soccer include test patterns such as the Illinois agility test, 5-0-5 test, Arrowhead test, or *t*-test, which all include straight sprinting and preplanned COD movement(s) [5,6]. These tests assess COD speed rather than agility, which incorporates a perceptual-motor component [7]. COD ability can also be masked by straight sprinting speed [6] and COD movements within a game are dissimilar because of flux in playing situations. Previous research has shown that the requirements for high-speed running, total distance, and CODs are position-specific [8–10], but little is known about the situational factors involved in COD maneuvers that could increase biomechanical demands during

the final foot contact. The analysis of biomechanical variables during CODs in a game would provide more specific information on the physical demands of gameplay and better guide training and testing tasks to match game requirements. Furthermore, most performance analysis studies in soccer involving COD biomechanics and wearables have been conducted with male participants, with only handful of studies on female youth athlete populations [11–14].

Wearable technology is used widely to monitor soccer players [15–18], including in professional leagues. However, the use of these data has been so far limited to estimating external load parameters or counts of specific movements (e.g., jumps, CODs) [19,20] with limited analysis of the biomechanical determinants of COD performance during gameplay matched with situational patterns. IMUs allow real-world data collection, but there is no current consensus on how this information should be used to support player development. There has been a call for more soccer-specific testing and monitoring, including a multidimensional approach for performance evaluations [21,22], which could be achieved in games with IMUs. IMUs could be utilized to provide information about the quality of not only whole-body movements, but also of individual segments during a COD movement [11]. In previous studies, this potential has been shown when analyzing walking, running, and postural control [23,24].

Laboratory-based studies have examined forces during COD movements and provided information about accelerations and ground reaction forces during the antepenultimate, penultimate, and final foot contacts [25–29]. The final foot contact (FFC) is the part of a COD that initiates the movement and allows the player to start accelerating in a new direction. Players who are able to enter the FFC phase with higher speed and greater ground reaction forces have been shown to perform better in traditional COD tests [25,27]. In addition, forces during the final step seem to be angle- and side-specific, meaning that the forces a player experiences while performing a COD will increase when the COD angle increases and alternate individually based on possible differences in leg dominance [30]. Laboratory-based studies with force plates have identified peak impact forces (braking and propulsive) as indicators of better performance [25,30]. Peak acceleration measured with IMUs has been shown to have the capacity to predict impact forces [31] and higher accelerations during on-field movements in team sports have been related to increased muscle damage and fatigue [32]. Based on a recent study, the vertical acceleration had the largest contribution to total acceleration during soccer games [33] and trunk-mounted IMUs have shown acceptable validity for measuring peak vertical and resultant forces [31].

Studies analyzing female populations during real-world challenges in evasion sports that require perception and reaction to other players movements are limited [12,14,34,35]. Considering larger incidence rates of anterior cruciate ligament (ACL) and ankle injuries in female youth players [2,36], there is a need to analyze biomechanical determinants of COD during games in female populations. Previous studies have shown that knee joint loading during the FFC is reduced when greater braking forces are applied during prior steps while decelerating [26,27,29]. Additionally, reduced decision times during tasks requiring reaction and the presence of a simulated defensive opponent have been shown to increase factors related to injury risk, such as knee abduction moments and knee valgus angles [37–39]. COD movement in soccer games is affected by approach speed, angle, anticipation, and positional requirements, which have not previously been studied thoroughly in female populations [30]. The lack of evaluation of demands for skill capacities in female athletes can lead to conclusions that disregard the multifactorial nature of high-speed movements [31].

To understand better game-specific demands for COD movement and to inform stakeholders working with youth female players for injury prevention or performance enhancement purposes, on-field evaluation of COD movements is warranted. Understanding the differences in on-field demands of different COD situations may lead to more soccer-specific training and testing methods that can support player development and injury prevention processes. For these reasons, the purpose of this study was to evaluate

the importance of situational features during CODs that predict IMU measured $PA_v$ during the final foot contact phase while turning in female youth soccer players. A secondary aim was to describe the factors that explained the greatest variance in $PA_v$ and thus the changing physical demands of COD movement.

## 2. Materials and Methods

### 2.1. Participants and Design

A convenience sample of one elite U16–U17 youth soccer team from a local soccer club in Calgary, Alberta, participated in this case series study. A total of 15 players played in the game and were included in the study. The mean age of players was 16.9 years (SD $\pm$ 0.36), height 163.9 cm (SD $\pm$ 6.4), body mass 59.7 kg (SD $\pm$ 9.5), and years played 10.6 (SD $\pm$ 2.1). Team recruitment occurred prior to the outdoor soccer season 2019 (May–October). Final participation was based on a signed mature minor consent form from each player and their parent/guardian. Ethical approval was granted by the University of Calgary Conjoint Health Research Ethics Board (REB19-0428).

### 2.2. Data Collection

Two trained research assistants equipped the players with a wireless triaxial IMU device (Shimmer 3 IMU $\pm$ 16 g, Shimmer Sensing, Dublin, Ireland, L4 level, in a pocket of an elastic strap) before the game. The devices were calibrated prior to data collection based on the manufacturer's recommendations (shimmersensing.com/product/shimmer-9dof-calibration). Players performed a general 20 min warm-up before the game, including running, skipping, dynamic stretches, passing drills, and shooting drills. Video materials were collected by videorecording the game with two 4 K cameras (Sony, FDR-AX53, 120 fps, Sony Corporation, Toronto, ON, Canada). The cameras covered opposite sides of the field (Figure 1). The player identification from video was performed based on jersey numbers of the players. Additionally, research assistants marked distinctive features of the players (e.g., hair and cleat color) on an information sheet that was double-checked while conducting the video analysis.

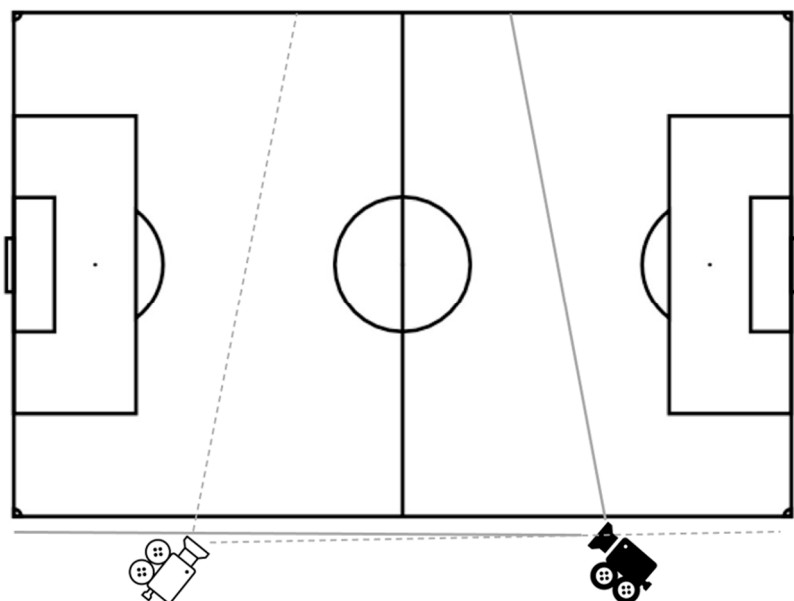

**Figure 1.** Video recording setup. (Camera 1 = white/dotted lines, camera two = black/solid lines).

The criteria for COD identification included that the player was running forward, changed direction while running, and continued the run without stopping towards the new direction (Figure 2). CODs that included backwards running, side-shuffling, or a combination of these with straight running were excluded. Situational patterns were

identified for each identified COD and included following features: (1) ball possession (player/team), (2) running speed (low/moderate or fast), (3) body contact with other players during COD movement, (4) turning side (right or left), (5) the angle of COD movement (90° cut, 135° cut, and 180° pivot turn), and (6) being challenged by an opposing player. Ball possession (player/team) was based on the ball possession definition presented by Link et al. [33]. Contact from other players to any body part was identified during or right before the COD maneuver and a challenge was identified as an action from opposing player that forced a perception–reaction response. Fast running speed was identified when the player was chasing the ball/opponent or being chased while having ball possession. Low/moderate speed was identified when a player was jogging and/or not pressing/being pressed. All COD movements, including field position (left/right, attacking/defensive half/field coordinates) for each COD and time of game (minute, first/second half) were identified using a soccer-specific tagging panel (Dartfish Live S). One reviewer (AMA), who has over ten years of experience in soccer coaching and game analysis, completed the video analysis.

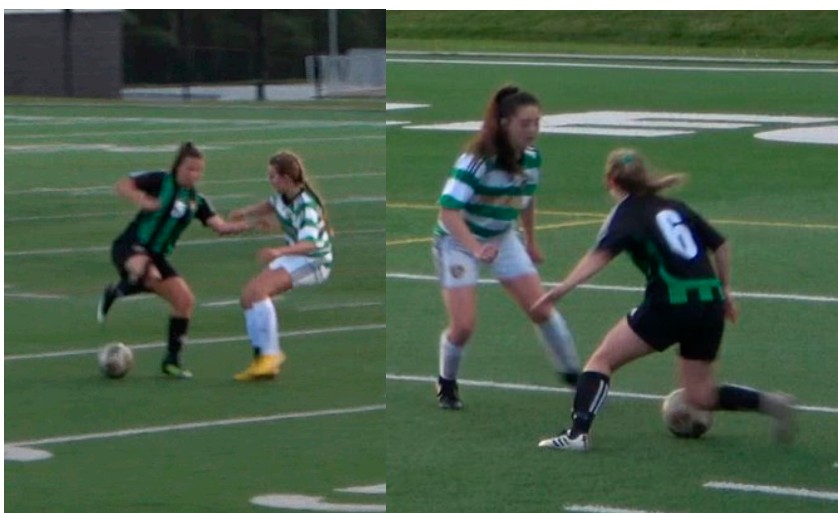

**Figure 2.** Players making 180° and 90° CODs with ball in the game.

Vertical plane raw peak accelerations ($PA_v$, m/s$^2$) for the final foot contact were derived from IMU data using a specific Matlab script (Version R2011b, MathWorks Inc., Natick, MA, USA) that synchronized the tagged situational datapoints with IMU data.

*2.3. Statistical Analysis*

Least absolute shrinkage and selection operator (LASSO) was used for the feature importance analysis. $PA_v$ during final foot contact was the outcome measure and categorical situational patterns, including playing position, player ID, and time of game (13 variables in total), were the predictor variables. Model performance was optimized by using a 10-fold cross-validation to obtain a model penalization parameter λ. Optimal value of λ was used to identify the key factors contributing towards $PA_v$ during COD. All statistical analysis was performed using R software, version 2022.12.0+353 [34], specifically the 'glmnet' package.

**3. Results**

LASSO regression identified four important features for the increase in $PA_v$ during CODs (Table 1). LASSO regression uses L1 regularization, which adds a penalty to the features; as a result, eight of the variables were shrunk to zero and were eliminated from the final model. The important features were contact before or during COD, running speed, challenge from opponent, and the angle of COD (Table 1).

**Table 1.** Important situational features affecting $PA_v$.

| Feature | β-Coefficient |
|---|---|
| Body contact before/during COD | 1.95 |
| Running speed | 2.78 |
| Challenge from opponent | 3.45 |
| COD angle | 7.15 |

COD = change of direction $PA_v$ = vertical peak acceleration.

Based on the β-coefficient values, COD angle appeared to have the greatest importance for Pav during COD movements, followed by challenge from opponent, running speed, and body contact before/during COD. The final LASSO model with these four parameters could be used to make predictions of accelerations on a new set of data. As the dataset in this case series study was small, all the data points were used to determine the best predictors for PAv, and the accuracy of the model in a test data set was not examined further.

Detailed information on the quantities of the CODs and different situational patterns is presented in Table 2. Playing time, half, or position on the field were not related to changes in $PA_v$ and were left out of the table.

**Table 2.** Quantities of different CODs with corresponding peak vertical accelerations (mean).

| Type of COD | Quantity | PAv (Mean, m/s²) |
|---|---|---|
| 180° | 107 | 41 |
| 135° | 85 | 33.66 |
| 90° | 144 | 25.2 |
| With ball | 138 | 32.1 |
| Without ball | 198 | 32.7 |
| Offensive | 187 | 32.5 |
| Defensive | 151 | 32.2 |
| Contact | 92 | 30.8 |
| No-contact | 245 | 37.5 |
| Right | 150 | 33 |
| Left | 186 | 31.5 |
| Fast | 104 | 30.5 |
| Low_moderate | 232 | 36.9 |
| Challenged | 173 | 35.1 |
| Not challenged | 163 | 29.4 |

COD = change of direction $PA_v$ = vertical peak acceleration.

## 4. Discussion

This case study evaluated the effects of situational features during COD movements on $PA_v$ in a youth female soccer game. The results of this study revealed higher $PA_v$ during COD being related to steeper turning angle, higher running speed, player being challenged, or having a contact before/during COD. The results of this research agree with the hypothesis set beforehand, which argued that impact forces during COD are angle- and speed-dependent, as shown in previous laboratory-based research. However, the results interestingly revealed additional factors that may increase accelerations during COD movement and potentially contribute to an increased neuromuscular demand.

Previous studies have shown angle- and speed-dependent changes in COD demands in laboratory conditions [25,30,35–37]. Sharper angles increased the COD demand measured by increased ground reaction forces [29], increased ground contact times, and decreased velocity profiles that reduced player COD performance [12]. These results support our findings that higher accelerations during CODs were associated with higher running speed and a steeper COD angle.

The results of this study show also that contact or challenge prior to or during a COD movement led to increased $PA_v$. Both having a contact prior or during COD and

being challenged by another player affected the players' ability to anticipate and achieve a more rapid perception–reaction. In an unanticipated COD task the muscle contraction strategy will change as well as the hip flexion angle, resulting in mechanical disadvantage, which most likely increases acceleration [38,39], which would explain the results seen in this study. Additionally, in unanticipated tasks, players have been shown to decelerate later, leading to higher accelerations during FFC [7,30,39]. Previous studies on anticipated and unanticipated tests have concluded that perception–action type activities need to be developed to ensure players' readiness for game demands. This is supported by the results of our study.

Previous studies have shown that greater horizontal forces during the FFC lead to faster 180° pivot turns [25] and faster approach velocities are beneficial for better performance [27,30]. In contrast, higher injury risk has been related to these same parameters [26]. Our study supported similar conclusions for on-field, unanticipated movements. This indicates the need for previously suggested modifications for strength training regarding more efficient and safe COD movement, but also introducing game-like elements to both training and testing, including varying running speeds and COD angles, pressing, and contacts.

This study is not without limitations, which should be considered when applying the results to practice. Only one game for one female youth team was included in the analysis. A longer follow-up and larger sample size would better capture the potential variation inherent in competitive soccer games. However, this study can provide valuable information for planning future studies for similar purposes and the results were supported by previous laboratory-based studies.

Coaches and sport science professionals should consider COD training and testing from a context-specific view and modifications to player testing are suggested. Including perception–reaction components, specifically opposition challenge and/or contact, to COD testing in addition with completing tasks with ball possession, would match better gameplay demands. In addition, COD training should include varying combinations of these elements for the purposes of player development and player evaluation. Consideration could also be given to the development of more sport-specific return-to-play testing protocols after injury.

## 5. Conclusions

This study showed that running speed, COD angle, contact, and challenge from opposing player were all related to higher $PA_v$ in the final step of COD movement during a soccer game. To better prepare youth soccer players for game demands, these factors should be included in COD training and testing.

**Author Contributions:** Conceptualization, A.-M.A., L.C.B., M.J.J., R.F. and K.P.; methodology, A.-M.A., L.C.B., M.J.J., R.F. and K.P.; software, A.-M.A. and L.C.B.; validation, A.-M.A. and L.C.B.; formal analysis, A.-M.A. and L.C.B.; investigation, A.-M.A.; resources, R.F. and K.P.; data curation, A.-M.A.; writing—original draft preparation, A.-M.A.; writing—review and editing, A.-M.A. and K.P.; visualization, A.-M.A.; supervision, K.P.; project administration, K.P.; funding acquisition, K.P. All authors have read and agreed to the published version of the manuscript.

**Funding:** This research received internal funding from the Faculty of Kinesiology, University of Calgary—Dean's Doctoral Scholarship and external funding from Sport Institute Foundation of Finland—Research Grant 20210058. Partial funding was provided by the NSERC Wearable Technology Research and Collaboration (WeTRAC) CREATE training program.

**Institutional Review Board Statement:** The study was conducted in accordance with the Declaration of Helsinki and approved by the Conjoint Medical Ethics Committee (REB19-0428).

**Informed Consent Statement:** Informed consent was obtained from all subjects involved in the study.

**Data Availability Statement:** The participants of this study did not provide consent for their data to be shared publicly.

**Acknowledgments:** The authors would like to thank Stephen Chaudhary, Meghan Critchley, Manraj Kang, Larissa Taddei, Stacey Sick, Haley Young, and Valeriya Volkova for their assistance with data collection.

**Conflicts of Interest:** The authors declare no conflict of interest.

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
