# Peer review of "Effects of Game-Specific Demands on Accelerations during Change of Direction Movements: Analysis of Youth Female Soccer"

_2673-7078, doi:10.3390/biomechanics3020021_

Round 1

Reviewer 1 Report

This short case-series manuscript presents a study on the use of an acceleration metric to assess change of direction performance in female soccer. The study presents in-field collected data for a generally underrepresented group of female athletes, for which the authors should be commended. However, important information on key concepts and variables is missing in the manuscript, the findings are based on a very limited sample size, and the implications of the findings could be further clarified, to strengthen the paper. I would, therefore, strongly encourage the authors to consider the attached comments.

Reviewer 2 Report

General comments

This paper provides knowledge regarding the mechanics of COD movements in female soccer players. I detected some flaws in this study, and I would like that, authors assessed them.

1st - regarding the scientific writing using the mechanics in football. I recommend for authors to deep search in specialized literature in football that analyzed the mechanical behavior on COD movements (i.e. Quoting the variables and bringing evidences about them).

2nd – I did not identify the real meaning of this research. What authors wants to answer?

3rd  – In the methods section, the inclusion of figures about the experimental protocol are required.

4th – In the results section, authors should explore more about the mechanical variables assessed and detected in the LASSO’s model.

5th – The discussion section should be based on each variable detected in LASSO’s model.

Specific comments

Title

 Is too long. I suggest authors to insert something relating to the main finding of this study.

Abstract

Line 19 – delete case series

Line 23-24 – Authors should expose briefly how was the experimental protocol.

Introduction

Line 54-55 – Who are the biomechanics determinants of COD. Insert a parenthesis quoting them and their specific reference.

Line 48-51 – This is not a plausible justification. Why authors want to study female soccers? Anthropometry factors  that can influence mechanical variables, for example? 

Line 76-80 – How was the meaning of this study? This is not clear on the introduction.

Methods

Line 91-94- A figure should be inserted here, showing the female soccer with respective IMU

Line 101-103 – Figure 1 should be improved. Authors should insert the female players on the field showing their positioning.

Line 104-121- I suggest authors to insert the variables description on a table.

Line 129-130- Where are theses dependent variables?

Results

135-141 – and about the others models generated by the Lasso. Could authors include the three last and their respective error value?

140-141- The present paper wants to be accepted in a biomechanical journal, please on this table correct the variable with their respective units.

Discussion

This section is too poor. The authors should discuss for each variable presented in LASSO’s model. Also, the authors should include a specific discussion for female soccers, as its was one of the justifications of this present study.

Line 157 – Authors used also neuromuscular data? If not, rewrite this sentence.

Line 169-173 – And measuring during various specific training sections?

Round 2

Reviewer 1 Report

I would like to thank the authors for considering and addressing the previous comments and feedback.

Reviewer 2 Report

Dear Editor, 

Many improvements have been done in the paper entitled “Effects of game specific demands on accelerations during change of direction movements: Analysis of youth female soccer”. The authors adopted all corrections. I recommend it for publication. 

My best regards.,